# Surgical Strategies to Dissect around the Superior Mesenteric Artery in Robotic Pancreatoduodenectomy

**DOI:** 10.3390/jcm11237112

**Published:** 2022-11-30

**Authors:** Kosei Takagi, Yuzo Umeda, Ryuichi Yoshida, Tomokazu Fuji, Kazuya Yasui, Jiro Kimura, Nanako Hata, Kento Mishima, Takahito Yagi, Toshiyoshi Fujiwara

**Affiliations:** Department of Gastroenterological Surgery, Okayama University Graduate School of Medicine, Dentistry, and Pharmaceutical Sciences, Okayama 700-8558, Japan

**Keywords:** robotic surgery, pancreatoduodenectomy, surgical approach, superior mesenteric artery, pancreatic cancer

## Abstract

The concept of the superior mesenteric artery (SMA)-first approach has been widely accepted in pancreatoduodenectomy. However, few studies have reported surgical approaches to the SMA in robotic pancreatoduodenectomy (RPD). Herein, we present our surgical strategies to dissect around the SMA in RPD. Among the various approaches, our standard protocol for RPD included the right approach to the SMA, which can result in complete tumor resection in most cases. In patients with malignant diseases requiring lymphadenectomy around the SMA, we developed a novel approach by combining the left and right approaches in RPD. Using this approach, circumferential dissection around the SMA can be achieved through both the left and right sides. This approach can also be helpful in patients with obesity or intra-abdominal adhesions. The present study summarizes the advantages and disadvantages of both the approaches during RPD. To perform RPD safely, surgeons should understand the different surgical approaches and select the best approach or a combination of different approaches, depending on demographic, anatomical, and oncological factors.

## 1. Introduction

The superior mesenteric artery (SMA) is an important anatomical landmark for pancreatoduodenectomy. The significance of artery-first approaches on clinical outcomes has been demonstrated in patients with pancreatic ductal adenocarcinoma (PDAC) undergoing a pancreatoduodenectomy [1,2]. Based on the growing evidence of the success of minimally invasive pancreatectomy as well as the shift from open to minimally invasive surgery, the indications for minimally invasive surgery for PDAC might be expanded [3,4]. To date, various surgical approaches for dissecting around the SMA have been reported in minimally invasive pancreatoduodenectomy, including anterior, posterior, left, and right approaches [5]. Although the right approach is commonly used in minimally invasive pancreatoduodenectomy, especially for robotic pancreatoduodenectomy (RPD) [5], the safety and feasibility of each approach remain controversial. Moreover, whether sufficient lymph node dissection around the SMA is possible using an appropriate approach is questionable. As appropriate lymph node dissection around the SMA may be associated with better prognosis in patients undergoing a pancreatoduodenectomy for PDAC [6], it is important to develop robot-specific surgical strategies to approach the SMA [7].

Herein, we present our surgical strategies for dissecting the SMA in RPD. Furthermore, a novel approach combining the left and right approaches during RPD is demonstrated.

## 2. Methods

### 2.1. Surgical Approaches to the SMA in RPD

Our standard protocol for RPD includes the right approach to the SMA (Figure 1a) [8,9], which was the only choice in most cases of pancreatic head and duodenum resection (Figure 1b). However, the left approach is used in patients with obesity, intra-abdominal adhesions, or malignant diseases requiring lymph node dissection around the SMA (Figure 1c). Therefore, we developed a surgical approach by combining the left and right approaches in RPD.

### 2.2. Dissection around the SMA Using the Right Approach

The anatomy around the SMA is shown in Figure 2. In the standard position, the left side of the SMA is difficult to dissect using the right approach (Figure 2a). However, the axis of the SMA could be rotated clockwise by approximately 90° by lifting the pancreatic head (Figure 2b). In the lifted position of the pancreatic head, the right approach allows dissection around the common trunk of the inferior pancreaticoduodenal artery (IPDA) and the first jejunal artery (J1A); however, circumferential dissection around the SMA could be difficult. An overview of the dissection area around the SMA using the right approach is illustrated in Figure 2c.

### 2.3. Dissection around the SMA Using a Combination of the Left and Right Approaches

To perform circumferential dissection around the SMA, we combined the left and right approaches. The left approach is suitable for dissecting the left side of the SMA (Figure 3a). In addition, the ligament of Treitz is easily divided, especially in patients with obesity. However, the transverse colon must be lifted to apply the left approach to the SMA.

To perform the left and right approaches, the transverse colon was first lifted cranially, and the left approach was adopted to expose the left aspect of the SMA (Figure 1c). In this step, the common trunk of the IPDA and the J1A was confirmed. However, only the origin of the J1A was divided using the left approach. Since division of the IPDA and vein by the left approach might lead to serious bleeding in minimally invasive surgery, we divided the IPDA and vein using the right approach. Thereafter, the transverse colon was repositioned and duodenal kocherization was performed. Once the jejunum was pulled into the right upper space, the right approach to the SMA was applied. The uncinate dissection along the SMA proceeds in a cranial direction by dividing the IPDA and several branches from the superior mesenteric vein in the lifting position of the pancreatic head (Figure 1b). Finally, circumferential dissection around the SMA was performed using a combination of left and right approaches in the RPD (Figure 3b and Appendix A).

## 3. Discussion

The appropriate surgical approach to safely perform RPD must be understood. In open pancreatoduodenectomy, various approaches to SMA have been reported, including the posterior, medial uncinate, mesenteric, left posterior, anterior, and superior approach, demonstrating the efficacy of the artery-first approach for early identification of resectability in patients with PDAC [1]. Furthermore, appropriate lymph node dissection around the SMA has been suggested for patients with PDAC [6]. Although it is important to understand the differences between open and robotic surgical approaches, only a few studies have reported surgical approaches to the SMA in RPD [5]. In this study, we describe robot-specific surgical strategies to approach the SMA during RPD.

The advantages and disadvantages of the right and left approaches during RPD are summarized in Table 1. Our first option is the implementation of the right approach, which is a common approach in a minimally invasive pancreatoduodenectomy [5]. Lymph node dissection around the SMA is possible, mostly on the right side of the SMA. Moreover, the tumor can be resected using the right approach without lifting the transverse colon. In RPD, it is technically difficult to create an adequate working space around the ligament of Treitz by lifting the transverse colon [10]. Therefore, the right approach is simple and feasible. In contrast, there are several technical difficulties associated with the right approach in patients with obesity, intraabdominal adhesions, or malignant diseases who require lymph node dissection around the SMA. This suggests that the left approach can overcome the disadvantages of the right approach, despite its disadvantages, such as a limited working space.

Considering both the advantages and disadvantages of the right and left approaches, we developed a surgical approach combining the left and right approaches in RPD. Starting the left approach by lifting the transverse colon allows easy dissection around the ligament of Treitz, even in patients with obesity and adhesions, and facilitates skeletonization at the left aspect of the SMA in patients with malignant diseases (Figure 4a). Circumferential skeletonization around the SMA is possible by proceeding with the correct approach (Figure 4b). The extent of lymph node dissection around the SMA in RPD should be determined based on surgical and oncological safety [11].

Evidence on the indications and advantages of an artery-first approach in RPD is limited, especially regarding surgical and oncological outcomes. However, surgeons should be familiar with different surgical approaches as well as the differences between open and robotic surgery. To perform RPD safely, we should select the best approach or a combination of different approaches depending on demographic, anatomical, and oncological factors.

## 4. Conclusions

The present study demonstrates our surgical strategies for dissecting the SMA during RPD. We believe that a surgical approach combining the left and right approaches could help in sufficient lymph node dissection around the SMA, especially in patients with PDAC undergoing RPD.

## Figures and Tables

**Figure 1 jcm-11-07112-f001:**
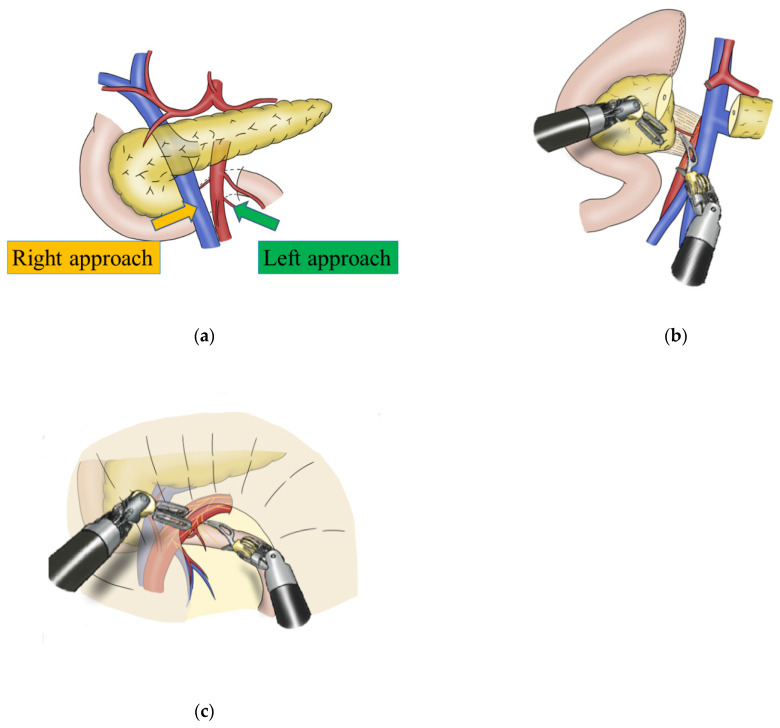
Surgical approaches to the superior mesenteric artery (SMA) in robotic pancreatoduodenectomy: (**a**) right and left approaches to the SMA; (**b**) right approach to the SMA; (**c**) left approach to the SMA.

**Figure 2 jcm-11-07112-f002:**
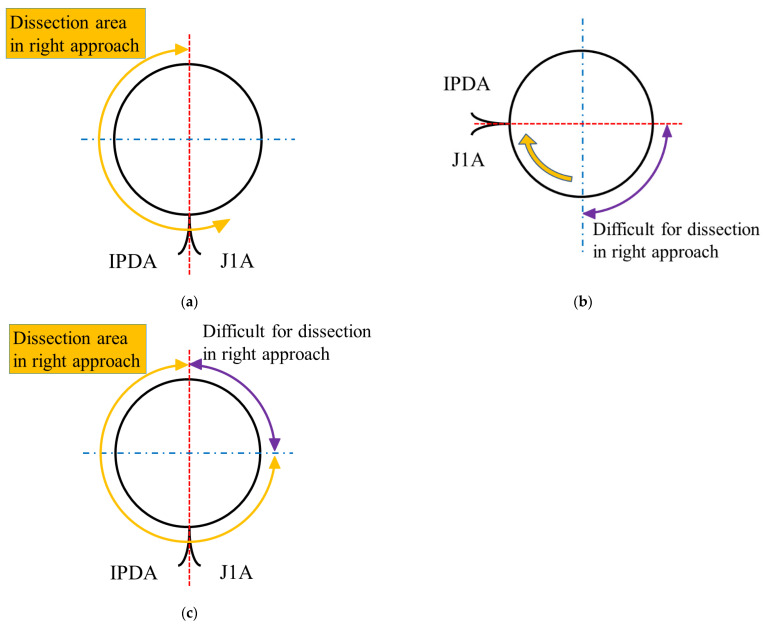
The anatomy around the superior mesenteric artery (SMA) during the right approach: (**a**) right approach is suitable for dissecting the right aspect of the SMA; (**b**) with the pancreatic head lifted, the axis of the SMA can be rotated clockwise approximately 90°, and the dissection area can be expanded; however, a difficult area for dissection still exists at the right side of the SMA (purple arrow); and (**c**) an overview of the dissection area around the SMA by the right approach. IPDA, inferior pancreaticoduodenal artery; J1A, first jejunal artery.

**Figure 3 jcm-11-07112-f003:**
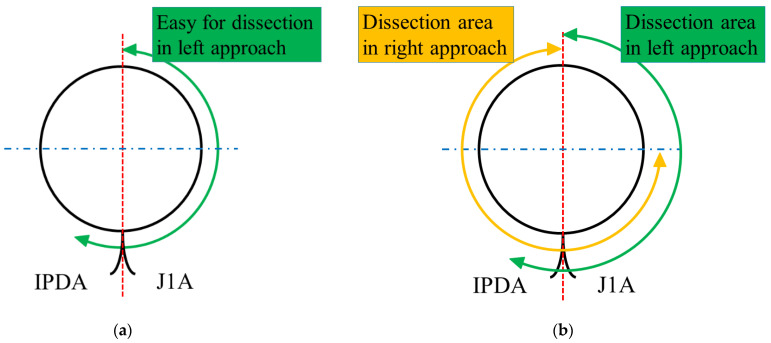
The anatomy around the superior mesenteric artery (SMA) during the left and right approaches: (**a**) the left approach is suitable for dissecting the left aspect of the SMA; and (**b**) the overview of circumferential dissection around the SMA using a combination of both approaches. IPDA, inferior pancreaticoduodenal artery; J1A, first jejunal artery.

**Figure 4 jcm-11-07112-f004:**
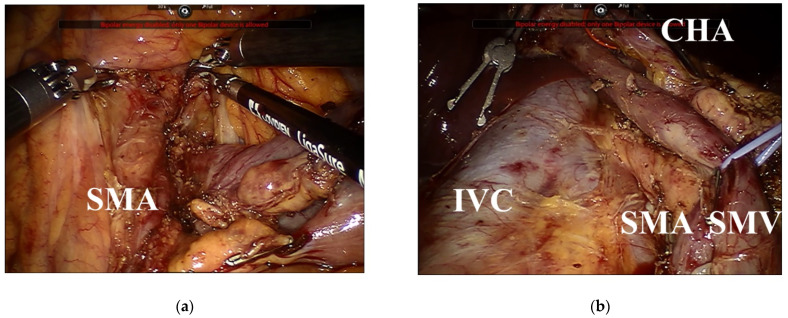
Lymphadenectomy around the superior mesenteric artery (SMA) in robotic pancreatoduodenectomy: (**a**) the left aspect of the SMA is exposed by the left approach; and (**b**) the right aspect of the SMA is skeletonized by the right approach. SMA, superior mesenteric artery; SMV, superior mesenteric vein; CHA, common hepatic artery; IVC, inferior vena cava.

**Table 1 jcm-11-07112-t001:** Our protocol to approaching the superior mesenteric artery in robotic pancreatoduodenectomy.

	**Indications**	**Advantages**	**Disadvantages**
Right approach	First option.	Commonly used in MIPD. Adequate working space. LN dissection mostly at the right side of the SMA. No need to lift the transverse colon.	Often difficult in patients with obesity. Difficult in patients with adhesions around the ligament of Treitz. Difficult to dissect at the left side of the SMA.
Left approach		LN dissection mostly at the left side of the SMA. Easy dissection around the ligament of Treitz, especially in patients with obesity.	Inadequate working space. Difficult to dissect at the right side of the SMA and uncinate process. Need to lift the transverse colon.
Left and right approach	Second option. Considered for patients with obesity, adhesions, and malignant diseases requiring LN dissection around the SMA.	Circumferential LN dissection around the SMA. Easy dissection around the ligament of Treitz.	Need to lift the transverse colon during the left approach.

MIPD, minimally invasive pancreatoduodenectomy; LN, lymph node; SMA, superior mesenteric artery.

## Data Availability

Not applicable.

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
