# Peer review of "Surgical Strategies to Dissect around the Superior Mesenteric Artery in Robotic Pancreatoduodenectomy"

_jcm, 2022, doi:10.3390/jcm11237112_

Round 1

Reviewer 1 Report

The manuscript " Surgical Strategies to Approach the Superior Mesenteric Artery in Robotic Pancreatoduodenectomy: A Nova approach" is an important and very interesting work in regard to robotic pancreatic surgery for malignancies of the pancreas.

The paper is well written, the approach is good understandable. Therefor, the  work is significant for surgeons who are interested in robotic surgery of the pancreas.  Once this promising approach has become established,  clinical data regarding the oncological outcome will be interesting.

Author Response

Thank you for your positive comments. We will evaluate the oncological outcomes in future studies.

Reviewer 2 Report

Dear authors

We read with great attention the manuscript entitled “Surgical Strategies to Approach the Superior Mesenteric Artery 2 in Robotic Pancreatoduodenectomy: A Novel Approach”.

This manuscript aims to investigate the interest of superior mesenteric first approach during robotic pancreatic head resection for PDAC. 

Artery first approach is characterized by early evaluation of involvement of the main arterial vasculature before irreversible surgical steps are performed as well as meticulous dissection of arterial planes and clearance of retropancreatic tissue. 

Moreover, it allows vascular control of pancreatico-duodenal arteries, and may decrease the operative blood loss which is also reported as prognostic factor. 

In summary, the SMA first approach is interesting because it allows to clear the right aspect of the SMA from the tumor before divided the pancreas which is the non-return point of pancreaticoduodenectomy (PD). The pancreatic isthmus transection should be performed only if the SMA is cleared and not before. 

Unfortunately, the technic described in the video and in this manuscript is not a superior mesenteric artery (SMA) first approach but could be described as a left to the right approach of the SMA in PD which remain an interesting technic as the pancreatic transection is perform 

Indeed, the authors described an extensive dissection from the left and the anterior aspect of the SAM first which is sometime involved when the tumor is located in the pancreatic uncus. This is a very nice approach and quite unusual and the robotic dissection is very well performed. This approach could also be referred to an extensive lymphadectomy (left aspect of the SMA) which is actually not systematically recommended in PD for PDAC because no oncological improvement have been identify and is associated with more postoperative morbidity.

The SAM first approach is a very regarding surgical technic and the caudal approach of minimally invasive surgery is probably very interesting but, in this video, the proximal and the right aspect of the SMA is dissected after pancreatic transection and this method is no a usual SMA first approach.

The manuscript writing, references and tables could support little improvement.

Author Response

Thank you for your insightful comments. We totally agree with reviewer 2’s comments regarding the concept of SMA-first approach. As reviewer 2 pointed out, the artery-first approach is characterized by early evaluation of the involvement of the main arterial vasculature before irreversible surgical steps are performed as well as meticulous dissection of arterial planes and clearance of retropancreatic tissue. In the present study, we would focus more on the surgical strategies to dissect around the SMA in robotic pancreatoduodenectomy, rather than “artery-first approach.” Therefore, we have changed the title as well as the aim of this study throughout the manuscript. Regarding the supplementary video, we have demonstrated our approach combining the left and right approaches during RPD, instead of “artery-first approach.”

Although the role of an extensive lymphadectomy might be controversial in PD for PDAC, further investigations should be conducted regarding the impact of lymphadenectomy on survival benefit.

Accordingly, we believe that surgeons should be familiar with various surgical approaches and select the best approach or a combination of different approaches depending on demographic, anatomical, and oncological factors.

Round 2

Reviewer 2 Report

Dear ahtohrs

thank you for your adquate corrections

Best regards